# Glycolipids Derived from the Korean Endemic Plant *Aruncus aethusifolius* Inducing Glucose Uptake in Mouse Skeletal Muscle C2C12 Cells

**DOI:** 10.3390/plants13050608

**Published:** 2024-02-23

**Authors:** Jong Gwon Baek, Do Hwi Park, Ngoc Khanh Vu, Charuvaka Muvva, Hoseong Hwang, Sungmin Song, Hyeon-Seong Lee, Tack-Joong Kim, Hak Cheol Kwon, Keunwan Park, Ki Sung Kang, Jaeyoung Kwon

**Affiliations:** 1KIST Gangneung Institute of Natural Products, Korea Institute of Science and Technology, Gangneung 25451, Republic of Korea; 223701@kist.re.kr (J.G.B.);; 2Department of YM-KIST Bio-Health Convergence, Yonsei University, Wonju 26593, Republic of Korea; ktj@yonsei.ac.kr; 3College of Korean Medicine, Gachon University, Seongnam 13120, Republic of Korea; 4Division of Bio-Medical Science and Technology, KIST School, University of Science and Technology (UST), Gangneung 25451, Republic of Korea

**Keywords:** *Aruncus aethusifolius*, glycolipid, glucose uptake, Korean endemic plant, molecular docking

## Abstract

*Aruncus* spp. has been used as a traditional folk medicine worldwide for its anti-inflammatory, hemostatic, and detoxifying properties. The well-known species *A. dioicus* var. *kamtschaticus* has long been used for multifunctional purposes in Eastern Asia. Recently, it was reported that its extract has antioxidant and anti-diabetic effects. In this respect, it is likely that other *Aruncus* spp. possess various biological activities; however, little research has been conducted thus far. The present study aims to biologically identify active compounds against diabetes in the Korean endemic plant *A. aethusifolius* and evaluate the underlying mechanisms. *A. aethusifolius* extract enhanced glucose uptake without toxicity to C2C12 cells. A bioassay-guided isolation of *A. aethusifolius* yielded two pure compounds, and their structures were characterized as glycolipid derivatives, gingerglycolipid A, and (2*S*)-3-linolenoylglycerol-*O*-β-d-galactopyranoside by an interpretation of nuclear magnetic resonance and high-resolution mass spectrometric data. Both compounds showed glucose uptake activity, and both compounds increased the phosphorylation levels of insulin receptor substrate 1 (IRS-1) and 5′-AMP-activated protein kinase (AMPK) and protein expression of peroxisome proliferator-activated receptor γ (PPARγ). Gingerglycolipid A docked computationally into the active site of IRS-1, AMPK1, AMPK2, and PPARγ (−5.8, −6.9, −6.8, and −6.8 kcal/mol).

## 1. Introduction

As technology has improved overall hygiene, food supply, and living standards, age-related disorders such as cardiovascular diseases, neurodegenerative diseases, and metabolic disorders have increased. In particular, approximately 422 million people suffer from diabetes and 1.5 million diabetes-related deaths occur each year according to the World Health Organization (WHO) [1]. Among them, type 2 diabetes mellitus (T2DM) accounts for 90%, potentially putting 541 million adults at risk of developing T2DM [2]. Although glucose is a critical component of metabolism and a primary source of energy for the body, its uptake and utilization are impaired in individuals with diabetes, resulting in hyperglycemia. Sustained hyperglycemia can damage blood vessels, nerves, and organs, leading to complications, such as heart disease, kidney problems, neuropathy, and blindness [3]. Therefore, maintaining blood sugar levels within a healthy range by improving glucose uptake may be an effective treatment strategy.

Insulin produced by pancreatic β-cells plays a pivotal role in maintaining glucose homeostasis. In general, insulin exerts its effects on target tissues such as the liver, fat, and skeletal muscle. However, a reduced capacity of insulin to elicit increases in glucose uptake and glycogen synthesis in target tissues is a common feature in T2DM patients [4]. This leads to a condition in which insulin action is defective and a given concentration of insulin fails to trigger glucose uptake signaling, referred to as insulin resistance (IR). The skeletal muscle is the largest contributor to glucose disposal and is the major site of IR in patients with T2DM [5]. Therefore, normal glucose uptake and metabolism in skeletal muscles are necessary for glucose homeostasis. The main goal of this study was to identify compounds with these related mechanisms.

Various therapeutic agents have been approved for the treatment of T2DM, with plant-derived molecules playing a critical role. Metformin, derived from *Galega officinalis*, has been used for more than 60 years [6]. To date, at least 1200 plant species have been used in traditional medicine by various communities owing to their anti-diabetic properties [7]. Hence, it is of great significance to identify various anti-diabetic compounds from these plants as well as their underlying mechanisms of action. *Aruncus* species belonging to the Rosaceae family have long been used in traditional folk medicines worldwide for their anti-inflammatory, hemostatic, and detoxification effects [8,9]. The well-known *Aruncus dioicus* var. *kamtschaticus* has long been used for multifunctional purposes in Eastern Asia, and recently, it was reported that its extract has antioxidant and anti-diabetic effects [10,11]. *A. aethusifolius* (H. Lev) Nakai is endemic to Korea; however, only a few studies have been conducted on it, including plant identification and proliferation studies [9,12]. Despite the lack of previous research on this plant, this species is also expected to have diverse biological activities such as anti-diabetic properties like other plants of the genus *Aruncus*. It may be worthwhile to look for biologically active molecules in this plant, especially anti-diabetic compounds. Here, we report the isolation, structural characterization, and biological evaluation of *A. aethusifolius* compounds.

## 2. Results and Discussion

### 2.1. Bioassay-Guided Isolation of Compounds

The 70% ethanolic extract of *A. aethusifolius* enhanced the glucose uptake in C2C12 cells. The biological evaluation results of the solvent-partitioned fractions suggested that the *n*-butanol fraction has the potential to contain biologically active compounds. Additional bioassay-guided fractionation showed that one subfraction significantly enhanced glucose uptake, and further isolation provided two biologically active compounds (Figure 1 and Figure 2A). The MTT assay showed that all the fractions had no toxic effects on C2C12 cells.

### 2.2. Structural Elucidation of Compounds

Compound **1** was isolated as a white amorphous powder, and its molecular formula was determined from HRESIMS data as C_33_H_56_O_14_, indicating six degrees of unsaturation. The mass fragmentation patterns at *m*/*z* 515 [M − 162 + H]^+^ and 353 [M − 324 + H]^+^ suggested that this compound is composed of one aglycone and two hexose units. Furthermore, the mass fragment at *m*/*z* 261 [M − 398 − H_2_O + H]^+^ indicated the presence of a glycerol unit and an additional functional group. The UV absorption patterns indicated that the compound lacks chromophores. A database search using this information indicated that the compound may be a monoacylglyceride. The NMR (Table 1) data showed the presence of typical signals for a di-galactosyl group (*δ*_C_ 105.3, 100.5, 74.7, 74.6, 72.6, 72.5, 71.5, 71.1, 70.2, 70.1, 67.8, 62.7) and a glycerol group (*δ*_C_ 72.1, 69.7, 66.6; *δ*_H_ 4.15, 4.00, 3.87, 3.67). The coupling constant of the anomeric proton suggested β-galactose moiety (*J* = 7.4 Hz, H-1′′), but another signal of the additional anomeric proton was overlapped. The location of the di-galactosyl group was determined by the HMBC correlations from H-1′′ (*δ*_H_ 4.25) to C-1′ (*δ*_C_ 72.1) and from H-1‴ (*δ*_H_ 4.86) to C-6″ (*δ*_C_ 67.8). The NMR data revealed a primary methyl group at *δ*_H_ 0.98 and six olefinic protons at *δ*_H_ 5.31–5.40, consistent with the typical signals of the linear unsaturated fatty acid, α-linolenic acid. The downfield shift of the proton lying on the C-3′ (*δ*_H_ 4.15) suggested the presence of the α-linolenic acid moiety at C-3′. This assumption was further supported by the HMBC cross-peak between H-3′ (*δ*_H_ 4.15) and C-1 (*δ*_C_ 175.5). The interpretation of the 2D NMR data and comparison with previous literature determined compound **1** as an *sn*-1 monoacylglycerol-type molecule, gingerglycolipid A (Figure 2) [13].

The molecular formula of compound **2** was determined to be C_27_H_46_O_9_, suggesting that this molecule lacks the monosaccharide of compound **1**. The mass fragmentation patterns at *m*/*z* 353 [M − 324 + H]^+^ and 261 [M − 398 − H_2_O + H]^+^ were similar to those of compound **1**, suggesting that this compound is composed of one aglycone and one hexose unit. The UV absorption patterns revealed the lack of chromophores. The NMR data (Table 1) showed only one anomeric signal (*δ*_C_ 105.3; *δ*_H_ 4.23) and an upfield shift of the carbon signal at C-6″ (*δ*_C_ 62.5), which suggested the galactosyl group was absent at C-6″. The interpretation of the 2D NMR data completed the structure and compound **2** was determined to be an *sn*-1 monoacylglycerol-type molecule, (2*S*)-3-linolenoylglycerol-*O*-β-d-galactopyranoside (Figure 2) [13].

The molecular formula of compound **2**-**1** was the same as that of compound **2**. The NMR data (Table 1) were also superimposable to those of compound **2**, except for the NMR data pattern of the glycerol group [**2**: *δ*_C_ 71.9 (C-1′), 69.6 (C-2′), 66.6 (C-3′), *δ*_H_ 3.92 (1 H, dd, *J* = 10.5, 5.1 Hz, H-1′a), 3.65 (1 H, dd, *J* = 10.5, 4.6 Hz, H-1′b), 3.99 (1 H, m, H-2′), 4.15 (1 H, m, H-3′); **2**-**1**: *δ*_C_ 68.8 (C-1′), 74.7 (C-2′), 61.7 (C-3′), *δ*_H_ 3.97 (1 H, dd, *J* = 10.8, 5.5 Hz, H-1′a), 3.75 (1 H, m, H-1′b), 5.05 (1 H, m, H-2′), 3.74 (1 H, m, H-3′)], suggesting that compound **2**-**1** was an *sn*-2 monoacylglycerol-type molecule, (2*S*)-2-linolenoylglycerol-*O*-β-d-galactopyranoside (Figure 2). This assumption was confirmed by the HMBC correlations from H-1′′ (*δ*_H_ 4.23) to C-1′ (*δ*_C_ 68.8) and from H-2′ (*δ*_H_ 5.05) to C-1 (*δ*_C_ 175.2). However, further analysis revealed that compound **2**-**1** rapidly interconverts with compound **2** through an acyl migration, which is a well-known reaction in monoacylglycerol. Their kinetics and equilibria depend on the structure, solvent, temperature, and pH. In our experiment, compound **2-1** was rapidly converted into a 90:10 mixture of 2:2-1 immediately after measuring the NMR data (pH 7), and we concluded that it was difficult to obtain compound **2-1** in a pure form [14]. The β-galactose moieties of compounds **1** and **2** were determined to possess a d-form (*t*_R_ 19.9 min) by comparing with standard d-galactose (*t*_R_ 19.9 min) and l-galactose (*t*_R_ 20.2 min) using the HPLC method performed in the previous study (Appendix A).

### 2.3. Effect of Compounds on Glucose Uptake in Skeletal Muscle Cells

Although three monoacylglycerols were identified in the effective fraction, the 1,2-diacyl shift suggested that only two compounds (**1** and **2**) could be reliably tested in the bioassay system. To evaluate the ability of compounds **1** and **2** to enhance glucose uptake, a 2-(*N*-(7-nitrobenz-2-oxa-1,3-diazol-4-yl) amino)-2-deoxyglucose (2-NBDG) glucose uptake assay was performed in C2C12 skeletal muscle cells. Both compounds induced glucose uptake at 25 and 50 µM (Figure 3A). Furthermore, both compounds did not show cytotoxicity below the effective concentration of 50 µM (Figure 3B).

### 2.4. Effect of Compounds on Protein Expression of p-IRS-1, IRS-1, p-AMPK, AMPK, PPARγ, and GLUT4

The effects of compounds **1** and **2** on protein expression in the C2C12 cells were also evaluated. Western blot analysis was performed to evaluate the effects of the compounds on the MAPK signaling pathway in C2C12 cells. The treatment with 50 μM of compounds **1** and **2** increased the phosphorylation levels of the insulin receptor substrate-1 (IRS-1) and 5′-AMP-activated protein kinase (AMPK) and the protein expression of the peroxisome proliferator-activated receptor γ (PPARγ) compared to untreated controls, whereas there were no meaningful results on the protein expression of the glucose transporter type 4 (GLUT4) (Figure 4).

### 2.5. Molecular Docking of Compounds on IRS-1, AMPK, and PPARγ

Molecular docking simulations using the representative compound **1** were conducted to reveal the potential protein-ligand interactions of the compound. The predicted binding scores for IRS-1, 5′-AMP-activated protein kinase catalytic subunit alpha-1 (AMPK1), 5′-AMP-activated protein kinase catalytic subunit alpha-2 (AMPK2), and PPARγ were found to be −5.8, −6.9, −6.8, and −6.8 kcal/mol, respectively. The specific interactions between compound **1** and its targets in the docking model were as follows.

IRS-1 formed hydrogen bonds with Arg213, Cys214, His216, Gln247, and His250 (Figure 5). The primary sugar moiety formed two hydrogen bonds with Arg213 at distances of 3.2 Å and 3.3 Å, and a single hydrogen bond with His216 at 3.0 Å. Additionally, the secondary sugar moiety established a water-mediated hydrogen bond with Arg213 and Cys214, and a direct hydrogen bond with His250 (2.6 Å), along with three hydrogen bonds with His216 (2.8 Å, 2.8 Å, 3.2 Å). The hydroxy ester of compound **1** also formed a hydrogen bond with Gln247 at a distance of 3.0 Å. In addition, hydrophobic interactions were observed between Phe222 and Leu254. The two sugar moieties were surrounded by hydrophobic interactions involving Gly215 and Phe222. Glu251, Leu254, and Arg258 participate in hydrophobic interactions [15]. These interactions played a pivotal role in stabilizing ligands within the binding site.

Compound **1** formed the most stable receptor-ligand complex (AMPK1) with a Vina score of −6.9 kcal/mol (Figure 6). Detailed structural analysis of the complex revealed that the primary sugar moiety was involved in hydrogen bond interactions in the kinase domain (KD) with the active-site residues Asp90 and Asn50. In addition, a single hydrogen bond was formed with Phe29 at a distance of 3.0 Å. The secondary sugar moiety created hydrogen bonds with Asp108 (3.0 Å), Asn110 (3.0 Å), and Asn111 (2.7 Å) in the carbohydrate-binding module (CBM). A shallow hydrophobic pocket formed between the CBM residue Lys31 and the KD residues Arg83 and His109. These structural features facilitated stable interactions, enabling the two sugar moieties to bind securely within the pocket. Similar to AMPK1, compound **1** established a stable receptor-ligand complex with a binding affinity of −6.8 kcal/mol (Figure 6). The hydroxy ester of compound **1** formed hydrogen bonds with the active site residue Val96 at a distance of 2.7 Å, as well as with Tyr95 (2.7 Å) and Ser97 (2.8 Å). In addition, the two sugar moieties of the compound formed hydrogen bonds with Gly167 and Leu22. Hydrophobic interactions were also observed between Leu146, Ala156, and Met164.

PPARγ showed a favorable binding affinity of -6.8 kcal/mol, similar to other targets (Figure 7). The active-site residue Arg316 formed hydrogen bonds with the two sugar moieties of compound **1**. The second active site residue, Ala327, formed hydrogen bonds with the second sugar moiety at distances of 2.9 Å and 3.2 Å. Additionally, Glu239 (2.5 Å), Arg371 (2.9 Å), Thr278 (3.4 Å), and Ile268 (2.4 Å) contributed further hydrogen bonds, collectively stabilizing compound **1** within the binding site. Consistent with the crystal structure [16], the docked complex exhibited hydrophobic interactions, which played a crucial role in stabilizing the ligand within the active site. Hydrophobic interactions involving Ala272, Trp305, Leu309, Ile310, Phe313, and Leu436 facilitated the stabilization of the compound to become stable within the binding site (Figure 7).

## 3. Discussion

Natural products are known to provide various health benefits, including nutritional value, antioxidant properties, and immune system-boosting effects. Natural products play an important role in the treatment of human diseases because plants synthesize diverse and potent secondary metabolites [17]. Hence, traditional remedies based on indigenous plants still dominate therapeutic practices worldwide, particularly for the treatment of diabetes. To date, at least 1200 plant species have been used as traditional remedies by various communities owing to their anti-diabetic properties [7]. In particular, metformin, derived from *Galega officinalis*, has long been used, and it is believed that many other undiscovered anti-diabetic molecules are present in plant sources [6]. *Aruncus* species have a long history of use in traditional folk medicine worldwide, with the most well-known being *A. dioicus* var. *kamtschaticus*, which has long been used in Eastern Asia for multifunctional purposes [8,9]. Although little research has been conducted on *A. aethusifolius*, it is expected to have important biologically active molecules, especially those with anti-diabetic properties.

The number of people with diabetes has increased dramatically in recent years, with type 2 diabetes accounting for 90% [2]. Hyperglycemia can damage various organs, leading to complications. Maintaining blood sugar levels within a healthy range and improving glucose uptake could be an effective treatment strategy. Anti-diabetic drugs comprise chemical or biochemical agents such as biguanides, sulfonylureas, thiazolidinediones, α-glucosidase inhibitors, glucagon-like peptide-1 receptor agonists, dopamine D2-receptor agonists, etc. [7]. However, these drugs can produce undesirable side effects such as kidney toxicity and gastrointestinal problems, including vomiting, indigestion, and diarrhea [18,19]. Therefore, the search for safer and more effective molecules from plant sources is highly important.

In the present study, glycolipid derivatives, gingerglycolipid A (**1**) and (2*S*)-3-linolenoylglycerol-*O*-β-d-galactopyranoside (**2**) were obtained from *A. aethusifolius*. To the best of our knowledge, the presence of these compounds in *Aruncus* species is reported for the first time in this work. In addition, only weak inhibitory effects of gingerglycolipid A on the growth of HepG2, AGS, HCT-15, and A549 cells have been reported, and there have been no reports on the anti-diabetic effects of these compounds [13]. Two compounds increased glucose uptake, which appeared to depend on the phosphorylation of IRS-1 and AMPK and the protein expression of PPARγ. In the molecular docking, the predicted binding scores for IRS-1, AMPK1, AMPK2, and PPARγ were found to be −5.8, −6.9, −6.8, and −6.8 kcal/mol, respectively. Specific analysis of protein-ligand interactions suggested that gingerglycolipid A could stably interact with these proteins within the binding site. In general, predicted molecular docking results and experimental results do not always agree well because a variety of factors can be involved, such as uptake, stability, and other interactions of the compounds, but in this study, the experimental results and molecular docking results agree well, suggesting the anti-diabetic potential of gingerglycolipid A. The evaluation of C2C12 cell glucose uptake and the involvement of various molecular factors, such as PPARγ, IRS-1, *p*-IRS-1, AMPK, and *p*-AMPK shed light on the intricate regulatory mechanisms that govern glucose homeostasis in skeletal muscle cells [20,21,22]. These factors play significant roles in glucose uptake and insulin sensitivity, and their interactions with, and effects on, metabolic health are significant. PPARγ is a transcription factor that plays an important role in adipogenesis and glucose metabolism [23]. It also enhances insulin sensitivity in skeletal muscle cells by promoting glucose uptake [24]. AMPK phosphorylation is a sign of activation and is often triggered by cellular energy depletion or various signaling pathways [25]. The activated AMPK promotes glucose uptake and maintains glucose homeostasis [26]. Although not directly involved in skeletal muscle glucose uptake, it is critical for maintaining overall glucose homeostasis by regulating insulin production [5]. The interactions between these factors highlight the complexity of the regulation of glucose uptake regulation [27]. Under healthy conditions, these molecules work together to ensure efficient glucose utilization by skeletal muscle cells. However, disturbances in this network can lead to insulin resistance and impaired glucose uptake, which are key features of metabolic disorders, such as T2DM. Hence, understanding the roles of PPARγ, IRS-1, *p*-IRS-1, AMPK, and *p*-AMPK in glucose uptake provides potential therapeutic targets for improving insulin sensitivity and glucose metabolism. We suggest that this compound has the potential to improve insulin sensitivity, glucose absorption, and glycemic control. Although further research is required to fully understand the underlying mechanisms and confirm these effects, these initial findings open exciting possibilities for the development of new approaches to manage and treat diabetes and related metabolic conditions. Moreover, it highlights the importance of exploring natural products that complement existing treatments for diabetes management.

## 4. Materials and Methods

### 4.1. General Experimental Procedures

Optical rotations were recorded using a PerkinElmer Model 343 polarimeter (Waltham, MA, USA). NMR data were acquired using a Bruker 500 MHz NMR spectrometer (Billerica, MA, USA), with chemical shifts referenced to the solvent peaks. HPLC chromatograms and ESIMS data were obtained using an Agilent 1200 system (Santa Clara, CA, USA) connected with a 6120 quadrupole MSD with a Phenomenex Luna C18 (2) column (5 μm, 150 × 4.6 mm, Torrance, CA, USA). HRESI-MS data were obtained using a Thermo Q-Exactive Orbitrap mass spectrometer via direct infusion (Waltham, MA, USA). Semi-preparative (Semi-prep) HPLC was carried out using the Gilson 321 HPLC system with a UV/VIS-151 detector (Middleton, WI, USA) and a YMC ODS-A column (5 μm, 250 × 20 mm, Tokyo, Japan). Flash column chromatography was performed using YMC reversed-phase silica gel (ODS-A, 12 nm, S-150 μm) and Merck silica gel (63–200 μm, 70–230 mesh, Darmstadt, Germany). Thin-layer chromatography (TLC) was performed using a Merck precoated silica gel F254 plate. The EZ-Cytox Cell Viability Assay Kit was obtained from ITSBIO (Seoul, Republic of Korea). Fetal bovine serum (FBS) was purchased from Invitrogen (Carlsbad, CA, USA). The primary antibodies for IRS-1 and GLUT4 and secondary antibodies for phosphorylated-IRS-1 (p-IRS-1) and AMPK, phosphorylated-AMPK (p-AMPK), and PPAR-γ were purchased from Cell Signaling (Danvers, MA, USA). All samples used in the biological evaluation were dissolved in dimethyl sulfoxide (DMSO).

### 4.2. Plant Material and Extraction

A whole *A. aethusifolius* plant was collected in May 2020 from the Hantaek Botanical Garden in Yongin-si, Republic of Korea, and authenticated by Taek Joo Lee (Hantaek Botanical Garden). A voucher specimen (HTS2020-0101) was deposited in the Herbarium of Hantaek Botanical Garden. The dried whole plant (375 g) was ground and extracted twice with 70% ethanol (3.8 L × 2) at 25 °C for 7 d to obtain a dark green extract (72.7 g).

### 4.3. Isolation and Structural Characterization of Compounds

The extract was dissolved in deionized water (0.8 L) and partitioned using *n*-hexane (1.2 L × 3) and *n*-butanol (1.2 L × 2). The *n*-butanol layer (13.9 g) was subjected to a silica gel column (45.0 × 5.0 cm, *n*-hexane/ethyl acetate/methanol, 3:1:0 to 0:1:1, each 1.8 L) to obtain 26 fractions (F1–F21). F21 (5.37 g) was subjected to flash C18 column separation (45.0 × 5.0 cm, methanol/water, 10:90 to 100:0, each 0.9 L) to obtain ten fractions (F21.1–F21.10). F21.9 (62.0 mg) was separated by semi-prep HPLC (acetonitrile/water, 11:14 in 120 min, flow rate 3.0 mL/min) at 210 nm to obtain gingerglycolipid A (1.9 mg, 40.2 min) and (2*S*)-3-linolenoylglycerol-*O*-β-d-galactopyranoside (1.4 mg, 108.3 min).

#### 4.3.1. Gingerglycolipid A (**1**)

White amorphous powder; [α]^23^_D_ +26.7 (*c* 0.01, MeOH); ^1^H and ^13^C NMR data (500 and 125 MHz, CD_3_OD), see Table 1; HRESIMS *m*/*z* 699.3544 [M + Na]^+^ (calcd for C_33_H_56_O_14_Na, 699.3568).

#### 4.3.2. (2*S*)-3-Linolenoylglycerol-O-β-d-Galactopyranoside (**2**)

White amorphous powder; [α]^23^_D_ +20.9 (*c* 0.01, MeOH); ^1^H and ^13^C NMR data (500 and 125 MHz, CD_3_OD), see Table 1; HR-ESI-MS *m/z* 537.3017 [M + Na]^+^ (calcd for C_27_H_46_O_9_Na: 537.3040).

### 4.4. Acid Hydrolysis

Acid hydrolysis was performed as described previously [28]. Compounds **1** and **2** (each 0.4 mg) were treated with 1 M of hydrochloric acid (400 μL) at 95 °C for 3 h and then partitioned with ethyl acetate (3 × 800 μL) to obtain aglycone moiety. The aqueous fractions were dried, evaporated to dryness, dissolved in pyridine (200 μL), treated with l-cysteine methyl ester hydrochloride at 60 °C for 1.5 h, and incubated with *O*-tolylisothiocyanate (25 μL) at 60 °C for 1.5 h. The reaction mixtures were analyzed by HPLC. Chromatographic separation was performed using an Agilent 1200 HPLC system equipped with a Phenomenex Luna C18 (2) column. The mobile phase consisted of water (A) and acetonitrile (B) containing 0.05% formic acid, at a flow rate of 0.7 mL/min. The gradient was set initially at 10% B and increased to 50% B for 30 min. The retention times were compared with those of the reacted mixtures (*t*_R_: d-galactose 19.9 min; l-galactose, 20.2 min).

### 4.5. Cell Culture

A mouse C2C12 skeletal muscle cell line was used. Dulbecco’s modified Eagle’s medium (DMEM) was supplemented with 10% FBS, antibiotics (penicillin/streptomycin), l-glutamine, and sodium pyruvate. The cells were incubated under conditions of 5% CO_2_ and 37 °C. The cells were centrifuged, and the collected cells were placed in the medium. The cells were then mixed and dispersed into single cells.

### 4.6. Cell Viability Assay

The cells were seeded in a 96-well plate at a density of 1 × 10^4^ cells/well in a volume of 100 μL per well and incubated for 24 h. They were treated with samples at various concentrations (6.25, 12.5, 25, and 50 μM) and incubated for 24 h. The control was treated with medium containing 0.5% (*v*/*v*) DMSO. After 24 h, 10 µL of EZ-Cytox was added to each well and incubated for 30 min. After the reaction, the absorbance was measured at 450 nm using a Biotek PowerWave XS microplate reader (Winooski, VT, USA). 100 μL of the culture medium used was mixed with 10 μL of EZ-Cytox and used as a blank.

### 4.7. Glucose Uptake Assay

C2C12 cells were cultured in high glucose DMEM medium supplemented with 10% fetal bovine serum and 100 μg/mL of streptomycin and penicillin at 37 °C in 5% CO_2_. The glucose uptake assay was performed as described previously. C2C12 cells were cultured in DMEM for 24 h to differentiate into myotubes, and the medium was changed to DMEM supplemented with 2% horse serum. After 5 d, the fully differentiated cells were treated with gliclazide or samples (6.25, 12.5, 25, and 50 μM) in serum-free high glucose DMEM containing fluorescence. The glucose uptake activity in the C2C12 myotubes was examined using a 2-NBDG assay kit (Sigma-Aldrich, St. Louis, MO, USA) according to the manufacturer’s instructions. The fluorescence intensities were measured at Ex/Em = 475 nm/550 nm using a Tecan microplate reader (Shanghai, China). Gliclazide was used as a positive control.

### 4.8. Western Blotting Analysis

The C2C12 cells (2 × 10^5^ cells/mL) were treated with different concentrations of the samples for 24 h, and cell extracts were prepared using radioimmunoprecipitation assay buffer (Cell Signaling Technology, Danvers, MA, USA) supplemented with a 1× protease inhibitor cocktail and 1 mM phenylmethylsulfonyl fluoride. Proteins (30 μg/lane) were separated by electrophoresis, transferred to polyvinylidene fluoride membranes, and bound to epitope-specific primary and secondary antibodies. Antibody binding was visualized using ECL Advance Western blot detection reagents (GE Healthcare, Buckinghamshire, UK) according to the manufacturer’s instructions and a LAS 4000 imaging system (Fujifilm, Tokyo, Japan).

### 4.9. Molecular Docking Analysis

Compound **1** was minimized using the AMBER ff14SB force field with the UCSF Chimera and converted into a pdbqt file using AutoDockTools 1.5.6 software. The three-dimensional structures for IRS-1, AMPK1, AMPK2, and PPAR**γ** were downloaded from a Protein Data Bank (PDB) with PDB accession numbers 1QQG, 6C9H, 3AQV, and 1FM9, respectively. The inhibitor bound to each target was used to define the ligand binding site for the molecular docking using AutoDock-Vina; IRS-1—Leu208, Met209, Ile211, Arg212, Cys214, His216, Ser217, Arg227, Ser228, and His250 [15,29], AMPK1—Phe29, Ile48, Asn50, Arg83, and Asp90 [30], AMPK2—Lys45, Tyr95, Val96, Asp103, Met164, and Leu146 [31], and PPARγ—Ala272, Trp305, Leu309, Phe313, Arg316, Leu326, Ala327, Val342, Cys432, and Leu436 [16]. The grid box dimensions and center were chosen by the binding site information with a default grid spacing of 0.375 Å. The lowest energy binding mode was selected for each target, and the interactions between the receptor and ligand were visually inspected using the PyMOL software (version 1.8.6.1).

## 5. Conclusions

A bioassay-guided isolation of *A. aethusifolius* led to the isolation of two glycolipid derivatives and the structures were characterized as gingerglycolipid A (**1**) and (2*S*)-3-linolenoylglycerol-*O*-β-d-galactopyranoside (**2**) using NMR and MS data. Both compounds showed glucose uptake activity, and both compounds increased the phosphorylation levels of IRS-1 and AMPK as well as protein expression of PPARγ. To verify these results in silico, a molecular docking simulation was performed to predict the binding orientation and affinity of gingerglycolipid A. Gingerglycolipid A docked computationally into the active site of IRS-1, AMPK, and PPARγ (−5.8, −6.9, −6.8, and −6.8 kcal/mol). A detailed analysis of the interactions between ligands and receptors, specifically identifying the major contributing factors, revealed the key residues and interaction patterns. This supports the results of the biological evaluation and provides valuable insights into the structural aspects of ligand binding.

## Figures and Tables

**Figure 1 plants-13-00608-f001:**
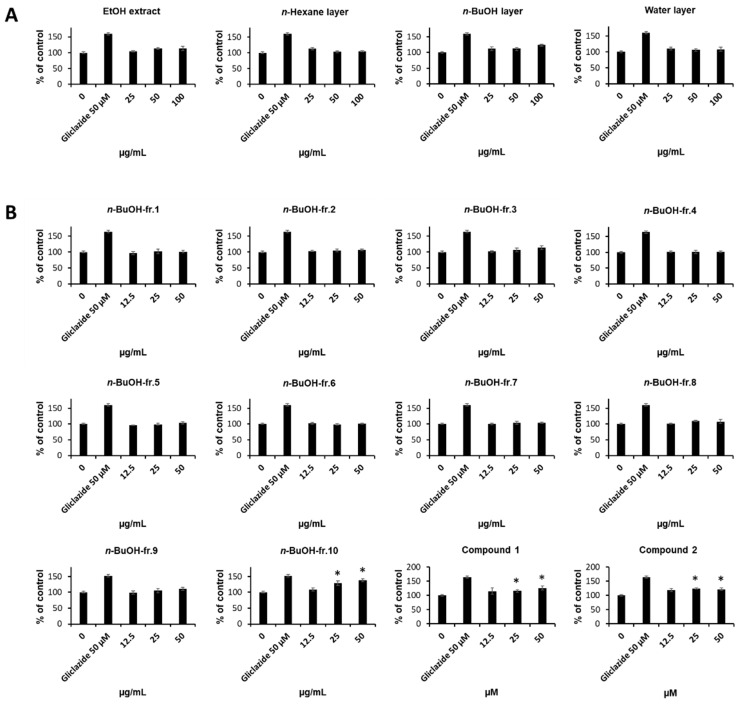
Effects of *A. aethusifolius* extract, solvent-partitioned fractions, and subfractions on the glucose uptake in C2C12 cells. Glucose uptake in C2C12 cells after 1 h incubation with (**A**) extract, solvent-partitioned fractions, and (**B**) subfractions, and 2-(*N*-(7-nitrobenz-2-oxa-1,3-diazol-4-yl) amino)-2-deoxyglucose (2-NBDG), assessed by glucose uptake assay. The indicated concentration of gliclazide was tested as a positive control for C2C12 cells. The data represent the mean ± S.E.M., *n* = 3, * *p* < 0.05 compared to the control.

**Figure 2 plants-13-00608-f002:**
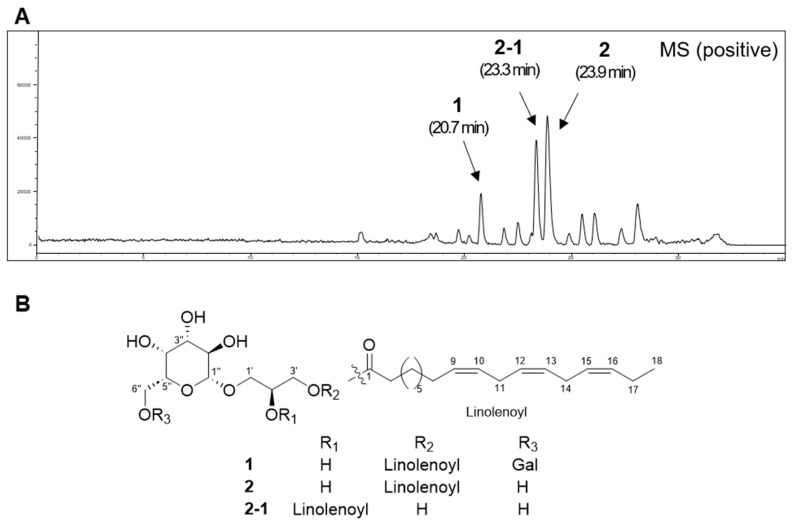
HPLC-MS analysis of active fraction of *A. aethusifolius* extract and the structures of isolated monoacylglycerols. (**A**) HPLC-MS chemical profiling of active fraction of *A. aethusifolius* extract monitored by the positive ion mode. (**B**) Structures of isolated monoacylglycerols.

**Figure 3 plants-13-00608-f003:**
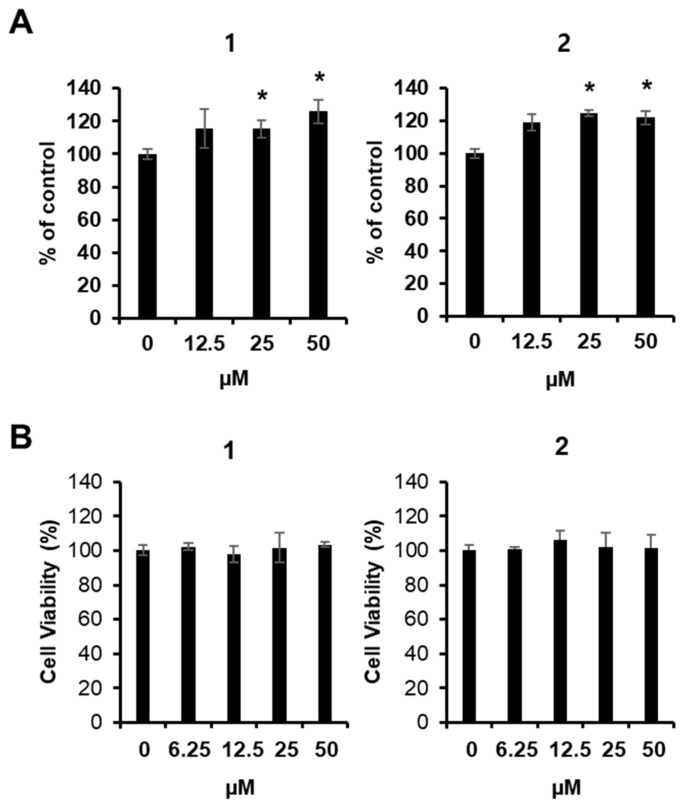
Effects of compounds on the glucose uptake in C2C12 cells. (**A**) Glucose uptake in C2C12 cells after 1 h incubation with compounds and 2-*N*-(7-nitrobenz-2-oxa-1,3-diazol-4-yl) amino)-2-deoxyglucose (2-NBDG), assessed by glucose uptake assay. (**B**) MTT assay results of the cell viability of C2C12 cells after 24 h treatment with compounds. Data represent the mean ± standard error of the mean (S.E.M.), *n* = 3, * *p* < 0.05 compared with the control.

**Figure 4 plants-13-00608-f004:**
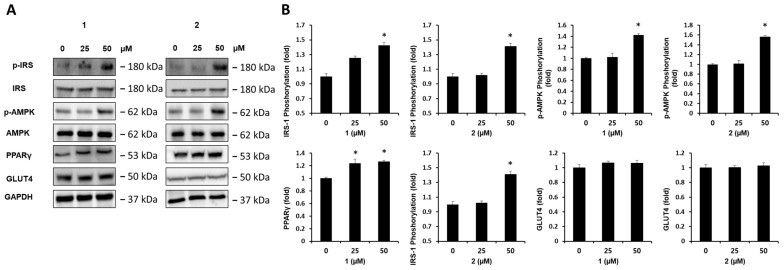
Effects of compounds **1** and **2** on the protein expression levels of phospho-insulin receptor substrate-1 (P-IRS-1), IRS-1, phosphorylated-5′-AMP-activated protein kinase (p-AMPK), AMPK, peroxisome proliferator-activated receptor γ (PPARγ), and glucose transporter type 4 (GLUT4). (**A**) *p*-IRS-1, IRS-1, *p*-AMPK, AMPK, PPARγ, GLUT4, and glyceraldehyde 3-phosphate dehydrogenase (GAPDH) in C2C12 cells treated or untreated with 25 and 50 μM of compounds **1** and **2**. (**B**) Each bar graph presents the densitometric quantification of Western blot bands. The data represent the mean ± S.E.M., *n* = 3. * *p* < 0.05 compared to the control.

**Figure 5 plants-13-00608-f005:**
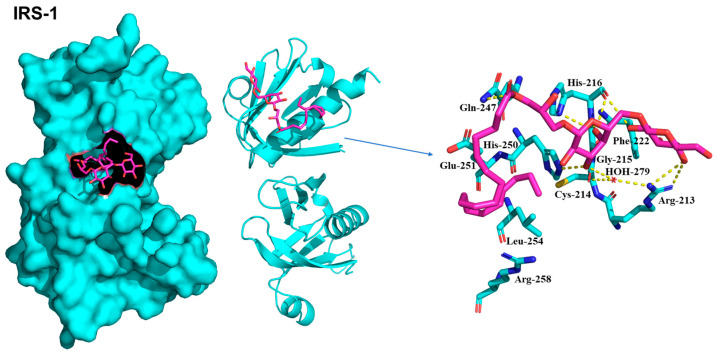
Docking model for IRS-1 (PDB ID:1QQG). The receptor and ligand are shown in cartoon (cyan color) and stick (magenta color), respectively.

**Figure 6 plants-13-00608-f006:**
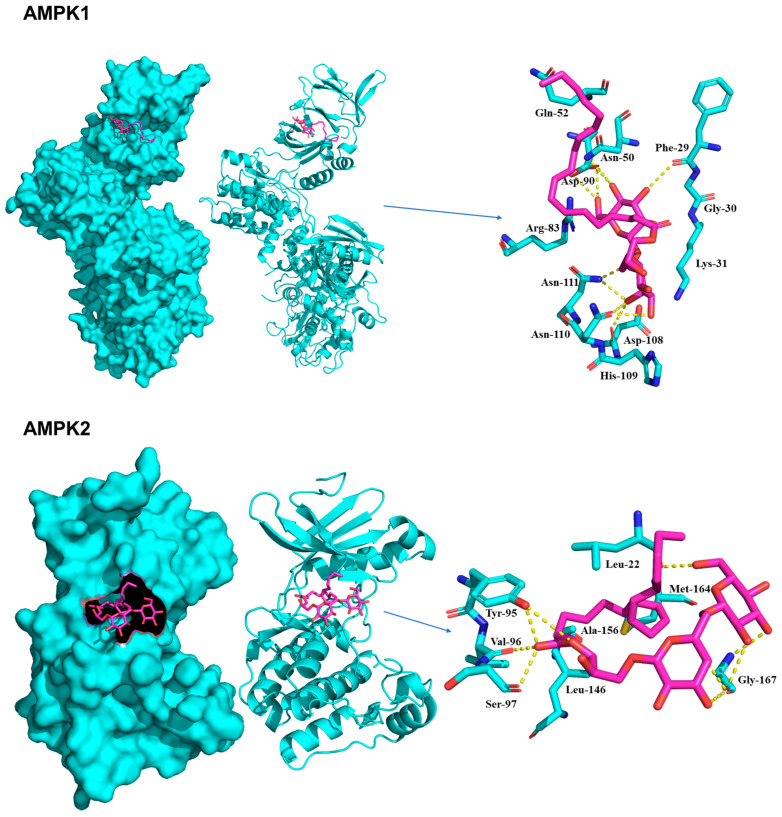
Docking models for AMPK1 and AMPK2 (PDB ID: 6C9H and 3AQV). The receptor and ligand are shown in cartoon (cyan color) and stick (magenta color), respectively.

**Figure 7 plants-13-00608-f007:**
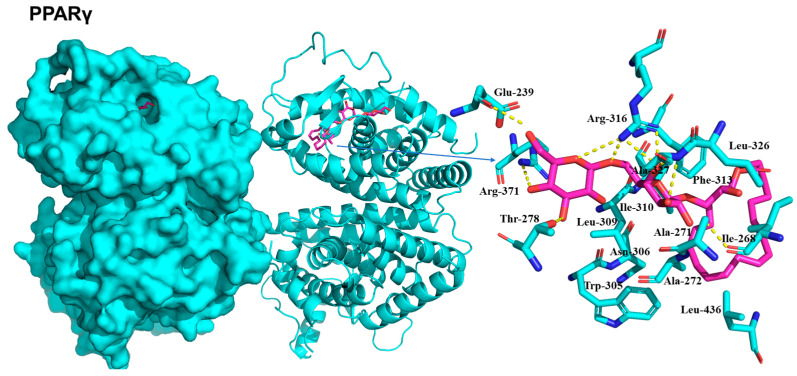
Docking model for PPARγ (PDB ID: 1FM9). The receptor and ligand are shown in cartoon (cyan color) and stick (magenta color), respectively.

**Table 1 plants-13-00608-t001:** NMR spectroscopic data for compounds **1**, **2**, and **2-1** in CD_3_OD.

No.	1		2		2-1	
	*δ* _C_	*δ* _H_	*δ* _C_	*δ* _H_	*δ* _C_	*δ* _H_
1	175.5		175.4		175.2	
2	34.9	2.36, t (7.5)	34.9	2.35, t (7.4)	35.1	2.36, t (7.5)
3	26.0	1.62, m	26.0	1.62, m	26	1.61, m
4	30.7 ^a^	1.34 *	30.7	1.34 *	30.7 ^a^	1.34 *
5	30.2 ^a^	1.34 *	30.2–30.3 ^a^	1.34 *	30.2 ^a^	1.34 *
6	30.2 ^a^	1.34 *	30.2–30.3 ^a^	1.34 *	30.2 ^a^	1.34 *
7	30.3 ^a^	1.34 *	30.2–30.3 ^a^	1.34 *	30.3 ^a^	1.34 *
8	28.2	2.07, m	28.2	2.09, m	28.2	2.08, m
9	131.1	5.31–5.40 *	128.3 ^b^	5.30–5.39 *	131.3	5.31–5.37 *
10	128.9 ^d^	5.31–5.40 *	128.9 ^b^	5.30–5.39 *	128.9 ^b^	5.31–5.37 *
11	26.4	2.81, m	26.4	2.81, m	26.4	2.81, m
12	129.2 ^d^	5.31–5.40 *	129.2 ^b^	5.30–5.39 *	129.2 ^b^	5.31–5.37 *
13	129.2 ^d^	5.31–5.40 *	129.2 ^b^	5.30–5.39 *	129.2 ^b^	5.31–5.37 *
14	26.5	2.81, m	26.5	2.81, m	26.5	2.81, m
15	128.2 ^d^	5.31–5.40 *	131.1 ^b^	5.30–5.39 *	128.3 ^b^	5.31–5.37 *
16	132.7	5.31–5.40 *	132.7	5.30–5.39 *	132.7	5.31–5.37 *
17	21.5	2.09, m	21.5	2.10, m	21.5	2.09, m
18	14.7	0.98, t (7.6)	14.7	0.98, t (7.6)	14.7	0.98, t (7.5)
1′	72.1	3.87, m	71.9	3.92, dd (10.5, 5.1)	68.8	3.97, dd (10.8, 5.5)
		3.67, m		3.65, dd (10.5, 4.6)		3.75, m
2′	69.7	4.00, m	69.6	3.99, m	74.7	5.05, p (5.5)
3′	66.6	4.15, m	66.6	4.15, m	61.7	3.74, m
1″	105.3	4.25, d (7.4)	105.3	4.23, d (7.6)	105.3	4.23, d (7.4)
2″	72.5 ^b^	3.53, m	72.6	3.54, m	72.4	3.52, m
3″	74.6	3.49, m	74.8	3.47, dd (3.4, 9.8)	74.9	3.46, dd (9.6, 3.3)
4″	70.1 ^c^	3.87, m	70.3	3.82, d (3.5)	70.3	3.82, d (3.3)
5″	74.7	3.75, m	76.8	3.52, m	76.8	3.50, m
6″	67.8	3.90, m	62.5	3.75, m	62.5	3.72, m
		3.67, m				
1‴	100.5	4.86 *				
2‴	70.2 ^c^	3.77, m				
3‴	71.5	3.74, m				
4‴	71.1	3.89, m				
5‴	72.6 ^b^	3.86, m				
6‴	62.7	3.71, m				

* Overlapped signals. ^a–d^ Values with the same superscript may be interchanged.

## Data Availability

Data are contained within the article and Appendix A.

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
