# Peer review of "Glycolipids Derived from the Korean Endemic Plant Aruncus aethusifolius Inducing Glucose Uptake in Mouse Skeletal Muscle C2C12 Cells"

_plants, 2024, doi:10.3390/plants13050608_

Round 1

Reviewer 1 Report

Comments and Suggestions for Authors

This paper describes the isolation of two compounds, both galactolipids and a secondary product.

There is at least one citation of closely related work that is not cited

Current Topics in Medicinal Chemistry, Volume 15, Number 11, 2015, pp. 1027-1034(8).

How do the current compounds compare to previously identified glycolipids in stimulating glucose uptake?

In Figure 2, the panels in A are not clearly enough labeled.  It is not obvious in the figure caption or text when compound 2-1 appears.  It is true that this rearrangement (between compounds of the general structures of 2 and 2-1) is facile but it is strongly condition dependent.  Do the authors know that 2-1 is not a natural product and were there any measures taken to reduce the acyl migration? 

The mechanism provided in Figure 3 would occur under basic conditions; what was the pH during the extraction and isolation?  Figure 3 is probably not required.

The analysis of NMR data is extremely generalized.  The paper text presents a range of chemical shifts consistent with the compounds identified but do not describe the analysis of the 2D NMR that would provide the connectivities actually supporting the identified natural products  (e.g., lines 114-117 on pg 4 “The NMR data showed the typical signal of the linear unsaturated fatty acid, α-linolenic acid. The interpretation of the 2D NMR data and comparison with previous literature determined compound 1…..”).  If the compounds are not new, their spectra should presented in the Supplementary Data (as they are currently) and the spectral data numerically correlated with literature data in a table in the manuscript body.  Otherwise, the table, supplementary data, and a rational structure elucidation should be given.

Line 112-113:  there are two anomeric protons in the digalactosyl group.  Which is being referred to, and what is the assignment of the other?

What the polarimeter used for (line 301)?  No optical rotations are given in the manuscript

It is not clear to the reviewer what the validation and ‘actual’ results are in the following comment on lines 273-4: “Although the predicted molecular docking results did not always agree perfectly with the actual results, the validation of both confirmed the anti-diabetic potential 274 of gingerglycolipid A.”  Please clarify.

It is not apparent from the plots in Fig. 5B that any effect can be claimed for changes in GLUT4 expression.

Author Response

Dear reviewer:

We appreciate your valuable suggestion regarding our manuscript entitled “Glycolipids derived from the Korean endemic plant Aruncus aethusifolius inducing glucose uptake in mouse skeletal muscle C2C12 cells”. We have carefully rechecked the manuscript according to the reviewer’s comments and we have corrected and responded to all the things the reviewer pointed out. We hope that the manuscript has been improved and this revised manuscript would be accepted for publication. Thank you very much for your consideration.

Reviewer #1

1. How do the current compounds compare to previously identified glycolipids in stimulating glucose uptake?

→ We thank the reviewer for the valuable comment. When we searched scifinder for compounds with high similarity (>85%) to gingerglycolipd A, there was only one previous study that indicated that glycolipids can induce glucose uptake (Verma et al., Biomed. Chromatogr., 2015, 29, 1675–1681). Since the main objective of the previous study was the development of HPLC method for the quantification of two glycolipids, the experiments were very simple and the results were very briefly described. While we fully understand the reasoning behind the reviewer's indication, we believe that the result of the previous study was not sufficient to compare with our result, so we have not described this.

2. In Figure 2, the panels in A are not clearly enough labeled. 

→ As the reviewer's indication, we have modified and added some information in figure 2A.

3. It is not obvious in the figure caption or text when compound 2-1 appears. It is true that this rearrangement (between compounds of the general structures of 2 and 2-1) is facile but it is strongly condition dependent. Do the authors know that 2-1 is not a natural product and were there any measures taken to reduce the acyl migration? The mechanism provided in Figure 3 would occur under basic conditions; what was the pH during the extraction and isolation? Figure 3 is probably not required.

→ We thank the reviewer for the valuable comment. When we made the extract using 70% ethanol (pH 7), compound 2-1 was already present in the extract and was also present during the subsequent fractionation. Therefore, our purpose was to obtain large peaks in the active fraction, so we obtained compounds 1, 2, 2-1 in high purity, and while compounds 1 and 2 were stable, compound 2-1 rapidly interconverted with compound 2 through an acyl migration. Previous studies suggested that this migration can occur under both acidic and basic conditions, and even during chromatography on silica gel (Richardson et al., Chem. Commun., 2017, 53, 1100-1103), making it difficult to control or measure. For this reason, we believed that this was not easily controllable in our environment, and we presented the phenomenon as it occurs. As the reviewer suggested, figure 3 usually occurs under basic conditions, so we have deleted figure 3 in the manuscript. Also, we have modified the description related to this in section 2.2 as below.

‘However, further analysis revealed that compound 2-1 rapidly interconverts with compound 2 through an acyl migration, which is a well-known reaction in monoacylglycerol. Their kinetics and equilibria depend on the structure, solvent, temperature, and pH. In our experiment, compound 2-1 was rapidly converted into a 90:10 mixture of 2:2-1 immediately after measuring the NMR data (pH 7), and we concluded that it was difficult to obtain compound 2-1 in a pure form.’

4. The analysis of NMR data is extremely generalized.  The paper text presents a range of chemical shifts consistent with the compounds identified but do not describe the analysis of the 2D NMR that would provide the connectivities actually supporting the identified natural products (e.g., lines 114-117 on pg 4 “The NMR data showed the typical signal of the linear unsaturated fatty acid, α-linolenic acid. The interpretation of the 2D NMR data and comparison with previous literature determined compound 1…..”).  If the compounds are not new, their spectra should presented in the Supplementary Data (as they are currently) and the spectral data numerically correlated with literature data in a table in the manuscript body.  Otherwise, the table, supplementary data, and a rational structure elucidation should be given.

→ In response to the reviewer's suggestion, we have incorporated a detailed description of the key 2D NMR correlations to enhance the clarity of the structure determination. Additionally, we have included a NMR table (Table 1) within the manuscript, highlighting the NMR data for compounds 1, 2, and 2-1.

5. Line 112-113: there are two anomeric protons in the digalactosyl group. Which is being referred to, and what is the assignment of the other?

→ As the reviewer suggested, we have added this information in Table 1 and section 2.2 as below.

‘The coupling constant of the anomeric proton suggested β-galactose moiety (J = 7.4 Hz), but another signal of the additional anomeric proton was overlapped.’

6. What the polarimeter used for (line 301)?  No optical rotations are given in the manuscript

→ As the reviewer suggested, we have added the polarimeter data in sections 4.3.1 and 4.3.2.

7. It is not clear to the reviewer what the validation and ‘actual’ results are in the following comment on lines 273-4: “Although the predicted molecular docking results did not always agree perfectly with the actual results, the validation of both confirmed the anti-diabetic potential 274 of gingerglycolipid A.”  Please clarify.

→ As the reviewer suggested, we changed this description in section 3 as below.

‘In general, the predicted molecular docking results are not always in good agreement with the actual results, we found a good match between the actual results and the molecular docking results, suggesting the anti-diabetic potential of gingerglycolipid A.’

8. It is not apparent from the plots in Fig. 5B that any effect can be claimed for changes in GLUT4 expression.

→ As the reviewer suggested, we have described the GLUT4 results as meaningless in section 2.4.

‘whereas there were no meaningful results on the protein expression of the glucose transporter type 4.’

Reviewer 2 Report

Comments and Suggestions for Authors

Western blot image need to show marker size.

Any biomolecular interaction data to support the molecular docking results, otherwise the molecular docking section looks out of place and without proof that the interaction actually happens.

Any microscopic data to show the uptake of 2NDB-glucose?

Any positive control used for the cell based assays?

Any pharmacological inhibitor has been used to support those pathways are involved for glucose uptake?

What solvent is used to dissolve the lipid?

Comments on the Quality of English Language

Overall ok

Author Response

Dear reviewer:

We appreciate your valuable suggestion regarding our manuscript entitled “Glycolipids derived from the Korean endemic plant Aruncus aethusifolius inducing glucose uptake in mouse skeletal muscle C2C12 cells”. We have carefully rechecked the manuscript according to the reviewer’s comments and we have corrected and responded to all the things the reviewer pointed out. We hope that the manuscript has been improved and this revised manuscript would be accepted for publication. Thank you very much for your consideration.

Reviewer #2

1. Western blot image need to show marker size.

→ As the reviewer suggested, the Western blot images have been revised to include marker sizes for better clarity and interpretation. The information regarding marker size will be included in the figure 4.

2. Any biomolecular interaction data to support the molecular docking results, otherwise the molecular docking section looks out of place and without proof that the interaction actually happens.

→ We thank the reviewer for the valuable comment. As the reviewer mentioned, biomolecular interaction analysis is necessary and we fully understand this. However, the purpose of our molecular docking experiments is to validate the results on protein expressions derived from the cellular experiments and to identify possible interactions between those proteins and the compound, suggesting the possibility that this type of compound can be utilized as an anti-diabetic compound. We believe that this type of compound is worthy of good utilization in the future, so we will conduct an optimization study of the compound and evaluate additional protein expressions and biomolecular interaction analysis of related mechanisms. If this goes well, we will also perform in vivo experiments. We hope that you will consider this positively.

3. Any microscopic data to show the uptake of 2NDB-glucose?

→ We thank the reviewer’s indication. However, in the present study, no microscopic measurements were made after 2NDB-glucose treatment. Absorbance was measured using the Glucose Uptake Cell-Based Assay Kit (Cayman, catalog number: 600470).

4. Any positive control used for the cell-based assays?

→ As the reviewer indicated, positive controls were utilized in the cell-based assays, and this information has been included in figure 1 and section 4.7.

5. Any pharmacological inhibitor has been used to support those pathways are involved for glucose uptake?

→ We thank the reviewer’s indication. We have not used pharmacological inhibitors. Although we would have liked to do this experiment, we have been unable to do this due to the time limit of the revision. Therefore, we will do this in further study, utilizing AMPK inhibitors such as compound C (Dorsomorphin) which were known to regulate glucose uptake by inhibiting AMPK.

6. What solvent is used to dissolve the lipid?

→ We dissolved all compounds, fractions, and extract in dimethyl sulfoxide. We have added this information in section 4.1.

Round 2

Reviewer 1 Report

Comments and Suggestions for Authors

Generally, the improvements over the initial version of this manuscript strengthened the paper.  The NMR data is more clearly presented and several of the issues around sample conditions were clarified.

There remain two issues that need consideration.

First, the revised statement "In general, the predicted molecular docking results are not always in good agreement with the actual results, we found a good match between the actual results and the molecular docking results, suggesting anti-diabetic potential of gingerglycolipid A." was an attempt to correct an earlier comment.  The issue is the word "actual".  What is an actual result?  Is a computational result actual? Assay? Binding study?  I think the authors mean "experimental" or "biological" rather than "actual".  There are a very large number of reasons why the magnitude of the docking interactions might not match with the biological assay (uptake of bioactive molecules, their stability, other interactions involving biological ligands with the targets chosen, effects on other aspects of glucose uptake).   I think the authors need to rewrite this sentence to convey clearly their point.

Second, in lines 415-418, it is written:  "The inhibitor bound to each target was used to define the ligand binding site  for the molecular docking using AutoDock-Vina; IRS-1 – _Leu208, Met209, Ile211, Arg212, 16 Cys214, His216, Ser217, Arg227, Ser228, and His250 [15,29], AMPK1 – _Asp90 [30], AMPK2  – _Val96 [31], and PPARγ _– _Arg316 and Ala327 [16]."

As written, this suggests the binding site interactions defined for AMPK1 and AMPK2 were simple one amino acid.  This either describes a binding site that is not meaningful, or the binding site definition should include a larger set of interactions.  Please revise accordingly.

Nitro in the chemical name of the assay reagent is not capitalized unless it is the first word in a sentence, as is generally the case for chemical names. 

Author Response

Dear reviewer:

We appreciate your valuable suggestion regarding our manuscript entitled “Glycolipids derived from the Korean endemic plant Aruncus aethusifolius inducing glucose uptake in mouse skeletal muscle C2C12 cells”. We have carefully rechecked the manuscript according to the reviewer’s comments. We hope that the manuscript has been improved and this revised manuscript would be accepted for publication. Thank you very much for your consideration.

Reviewer #1

Generally, the improvements over the initial version of this manuscript strengthened the paper.  The NMR data is more clearly presented and several of the issues around sample conditions were clarified. There remain two issues that need consideration.

1. First, the revised statement "In general, the predicted molecular docking results are not always in good agreement with the actual results, we found a good match between the actual results and the molecular docking results, suggesting anti-diabetic potential of gingerglycolipid A." was an attempt to correct an earlier comment. The issue is the word "actual". What is an actual result? Is a computational result actual? Assay? Binding study? I think the authors mean "experimental" or "biological" rather than "actual". There are a very large number of reasons why the magnitude of the docking interactions might not match with the biological assay (uptake of bioactive molecules, their stability, other interactions involving biological ligands with the targets chosen, effects on other aspects of glucose uptake). I think the authors need to rewrite this sentence to convey clearly their point.

→ We thank the reviewer for the valuable comment. As the reviewer suggested, we revised this sentence as below.

“In general, predicted molecular docking results and experimental results do not always agree well because a variety of factors can be involved, such as uptake, stability, and other interactions of the compounds, but in this study, the experimental results and molecular docking results agree well, suggesting the anti-diabetic potential of gingerglycolipid A.”

2. Second, in lines 415-418, it is written: "The inhibitor bound to each target was used to define the ligand binding site for the molecular docking using AutoDock-Vina; IRS-1 – _Leu208, Met209, Ile211, Arg212, 16 Cys214, His216, Ser217, Arg227, Ser228, and His250 [15,29], AMPK1 – _Asp90 [30], AMPK2 – _Val96 [31], and PPARγ _– _Arg316 and Ala327 [16]."

As written, this suggests the binding site interactions defined for AMPK1 and AMPK2 were simple one amino acid. This either describes a binding site that is not meaningful, or the binding site definition should include a larger set of interactions. Please revise accordingly.

→ We thank the reviewer for the valuable comment regarding the description of the binding site residues of AMPK1 and AMPK2. In the previous version of the manuscript, we presented the key amino acids (hot-spots) crucial for the protein-ligand interactions. In the revised manuscript, we have expanded upon this by providing all the binding residues involved in the binding sites of AMPK1, AMPK2, and PPARγ.

“The inhibitor bound to each target was used to define the ligand binding site for the mo-lecular docking using AutoDock-Vina; IRS-1 – Leu208, Met209, Ile211, Arg212, Cys214, His216, Ser217, Arg227, Ser228, and His250 [15,29], AMPK1 – Phe29, Ile48, Asn50, Arg83, and Asp90 [30], AMPK2 – Lys45, Tyr95, Val96, Asp103, Met164, and Leu146 [31], and PPARγ – Ala272, Trp305, Leu309, Phe313, Arg316, Leu326, Ala327, Val342, Cys432, and Leu436 [16].”

3. Nitro in the chemical name of the assay reagent is not capitalized unless it is the first word in a sentence, as is generally the case for chemical names.

→ We thank the reviewer for the valuable comment. We revised all the chemical names properly.

Reviewer 2 Report

Comments and Suggestions for Authors

Authors have addressed most of my concerns. I do not have further questions.

Comments on the Quality of English Language

Overall English is fine

Author Response

We thank the reviewer for the helpful comment.